# A Perspective on Recent Advances in Piezoelectric Chemical Sensors for Environmental Monitoring and Foodstuffs Analysis

**Tatyana A. Kuchmenko [1] and Larisa B. Lvova [2,\*]**

[1]  Department of Ecology and Chemical Technology, Voronezh State University of Engineering Technologies, Revolution Avenue 19, Voronezh 394000, Russia
[2]  Department of Chemical Science and Technologies, University "Tor Vergata", via della Ricercha Scientifica 1, 00133 Rome, Italy
[\*]  Correspondence: Larisa.Lvova@uniroma2.it

**Abstract:** This paper provides a selection of the last two decades publications on the development and application of chemical sensors based on piezoelectric quartz resonators for a wide range of analytical tasks. Most of the attention is devoted to an analysis of gas and liquid media and to industrial processes controls utilizing single quartz crystal microbalance (QCM) sensors, bulk acoustic wave (BAW) sensors, and their arrays in e-nose systems. The unique opportunity to estimate several heavy metals in natural and wastewater samples from the output of a QCM sensor array highly sensitive to changes in metal ion activity in water vapor is shown. The high potential of QCM multisensor systems for fast and cost-effective water contamination assessments "in situ" without sample pretreatment is demonstrated.

**Keywords:** quartz crystal microbalance (QCM) sensors; chemical sensors; environmental monitoring; food quality and safety assessment

---

## 1. Introduction

Chemical sensors represent one of the most significant tools of analytical chemistry. Relatively simple in preparation and application and inexpensive, these devices allow for the identification and determination of substances in their gaseous and liquid phases and may function in automatic and remote modes while being implemented in various technological processes. Moreover, they have found wide applications in medicine, agriculture, and environmental monitoring [1,2].

It is hard to overestimate the worldwide popularity of research devoted to the development and application of different types of sensors. Recently, the design and application of sensors have evolved into an independent branch of technology and measuring equipment. The growth of automation and the development of smart sensor and intelligent domotics concepts have increased the requirements imposed on modern sensors and detectors. In particular, special attention is given to sensors' mechanical robustness, chemical inertness, parameter independence on external conditions (positioning in space, temperature, and pressure), selectivity, sensitivity, possibility of automation, size, and cost. Previously, several comprehensive books and reviews on chemical sensors and sensor array principles and applications were published [3–13]. Nowadays, oscillatory-based sensory systems cover a significant part of the total number of sensors employed [1,14–17]. Among them, sensing systems with lumped parameters (rigid bodies) or systems with distributed (continuous) parameters can be distinguished (Figure 1).

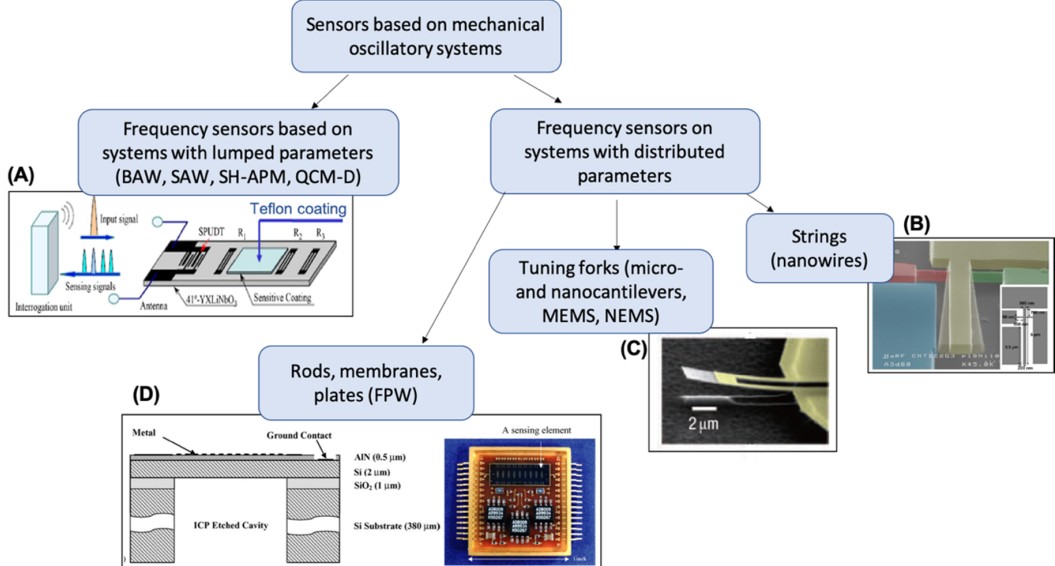

**Figure 1.** Classification of oscillatory-based sensors. (**A**) Schematic diagram of the wireless surface acoustic wave (SAW) $CO_2$-gas sensor system, reprinted from Reference [18]; (**B**) Scanning electron microscope (SEM) image of an in-plane silicon nanowire piezoresistive resonator, reprinted from Reference [19]; (**C**) SEM micrograph of an SiC nanocantilever with dimensions of 10 um × 2 um and 70 nm thick, reprinted from Reference [20]; (**D**) cross-section diagram of a flexural plate wave (FPW) resonating membrane structure and an image of chemical sensor arrays with differential outputs (uCANARY), reprinted from Reference [21]. On figure—BAW: bulk acoustic wave; SH-APM: shear horizontal acoustic plate mode sensor; QCM-D: quartz crystal microbalances with energy dissipation control.

## 2. Piezoelectric Sensors Based on Gravimetric Resonant Devices

### 2.1. Design and Operation Principle

Gravimetric or mass-sensitive devices transform the mass change at a specially modified surface into a change of a property of the support material. The mass change is caused by accumulation of the analyte. The piezoelectric effect lies in the principle of such a device response. This effect consists of the generation of electric dipoles inside certain elastic anisotropic crystals when subjected to mechanical stress. Piezoelectrically excited bulk acoustic wave (BAW) resonators, quartz crystal microbalances with control of energy dissipation (QCM-D), and surface acoustic wave (SAW) resonators are still the most widely used in analytical practice [15–17,22,23]. A flexural plate wave (FPW) and a shear horizontal acoustic plate mode (SH-APM) have also been employed in chemical sensor development [19,21]. The beginning of the current century was characterized by a new direction in instrument-making, micro- and nanoelectromechanical systems development, MEMS, and NEMS [16,20,24–27]. Moreover, a new microgravimetry instrument, an electrochemical quartz crystal, has been demonstrated to be particularly promising for studying the viscoelastic properties of redox-active thin films and conductive polymers [28].

A typical example of a BAW device is the thickness shear mode (TSM) resonator, which is also widely referred to as a quartz crystal microbalance (QCM). A QCM is generally made of a thin disk of quartz crystal fixed between two circular metallic (most often golden) electrodes with fundamental frequencies between 5 and 50 MHz (Figure 2A). When an electric field is applied between the electrodes, the crystal is mechanically deformed and oscillates with a fundamental frequency. As the electrodes are attached to either side of the crystal, the wave produced travels though the bulk of the material, as illustrated in Figure 2B. There are different cuts of quartz crystals, which permits varied desired wave propagation features: among them, the AT-cut (35°15′ inclination in the *y*–*z* plane) is the most widely used.

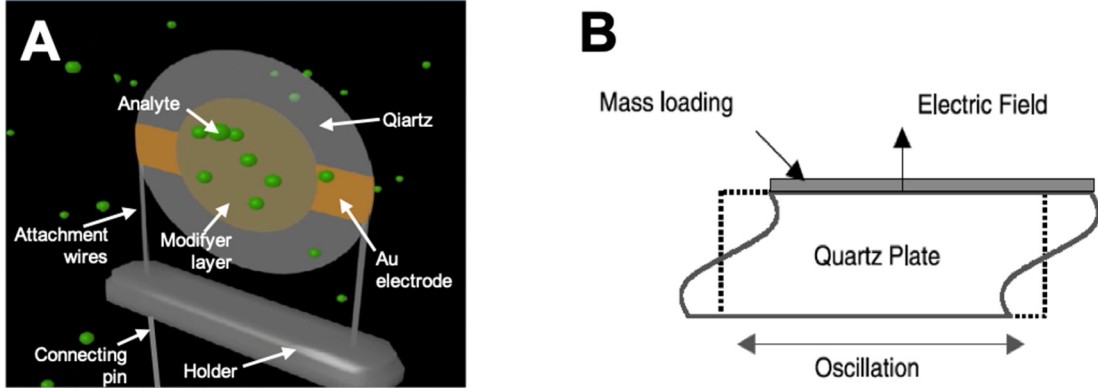

**Figure 2.** (**A**) 3D model of QCM sensor in mode of vapor sorption; (**B**) schematic presentation of BAW oscillation.

Any layer added to the quartz crystal, which does not dampen the oscillation, can be treated as added thickness, causing a change in frequency. Even a tiny mass change of the sensor causes a measurable frequency shift. For a standard sensor, a 1-Hz change can relate to a 1-ng mass change. If soft polymers are used to coat the electrode, it is possible for them to have viscoelastic coupling to the acoustic wave: in these circumstances, the frequency shift cannot be entirely due to mass change. The QCM mass sensitivity can be expressed by the Sauerbrey equation [29]:

$$\Delta f = -\frac{2f_0^2}{A \cdot \sqrt{\rho_q \cdot \mu_q}} \cdot \Delta m, \tag{1}$$

where $\Delta f$ is the change in frequency of the quartz crystal in Hz; $f_0$ is the fundamental frequency of the quartz crystal (Hz); $\Delta m$ is the mass change of material deposited or sorbed onto the crystal in g; $A$ is the area between electrodes in cm$^2$; $\rho_q$ is the density of quartz; $\mu_q$ is the shear modulus of quartz for an AT-cut crystal; and $v_q$ is the transverse wave velocity in quartz in m/s.

In SAW devices, comb-shaped interdigital metallic transducers (IDTs) are patterned over the piezoelectric material by a photolithographic or other process. When an alternate current is applied to IDTs, the surface acoustic waves are propagated and guided along the surface of an elastic layer, with most of the energy density confined to a depth of about one wavelength below the surface. The selection of a different piezoelectric material and appropriate crystal cut results in a shear horizontal acoustic wave in shear horizontal acoustic plate mode sensor (SH-APM) resonators. In quartz crystal microbalances with energy dissipation control (QCM-D) devices, simultaneous measurements in resonance frequency ($\Delta F$) and energy dissipation ($\Delta D$) changes are performed. For this purpose, the driving power applied to the sensor crystal is switched off periodically, and the decay of damped oscillation is recorded as the result of an analyte adsorption or other structural changes [30]. In this way, a QCM-D resonator may provide structural information on the viscoelastic properties of liquids and adsorbed films and can be employed for coating property characterization.

Among those mentioned above, each type of resonator has a number of advantages and limitations. For the manufacture of chemical sensors, transducers based on BAW–piezo resonators remain the most popular. Moreover, in the past years, there has been a tendency to substitute the individual selective piezo-microbalances with multisensor systems based on various sensing elements. Nowadays the number of publications devoted to application of such devices in traditional areas of analysis (mainly for food systems), but also for less studied earlier objects in environmental monitoring, biological tests, technological processes, studying biological and other reactions at the molecular level is expanding.

## 2.2. Piezosensors Optimization Strategies

To obtain a chemical sensor from a piezoelectric resonator (as an oscillatory system), it is necessary to modify resonator electrodes through specific reagents that are capable of altering their properties (in particular, mass) due to selective reactions with detectable components (ions, molecules, their fragments or clusters) in the analyzed environment. In order to properly develop a new analytical method, the following three factors should be considered: (i) proven theoretical backgrounds, including a description of the operation principles and prediction of system properties when externally exposed; (ii) a well-established method of measuring device and gadget manufacturing; and (iii) prompt practical solutions and analysis techniques. Despite initial interest, new mathematical models describing the functioning of piezoelectric masses with different coatings on electrodes have not received further development, even if the significant modifications may occur inside coatings as far as inside analyzed gaseous or liquid objects during the sensing event. This can be associated with the displacement of traditional transducers with thin-film acoustic sensors and with the sufficient versatility of the Sauerbrey model [29], when the resonators are loaded with thin films and when analytes are detected in dilute solutions. On the contrary, new approaches that enhance the useful response of piezoelectric balances and increase their resolution by modifying film coatings on a quartz plate or electrodes with gold nanoparticles [31], chitosan [32], or particles with magnetic properties [33] through the formation of biomimetic silica, peptide silaffin (silaffin peptides) [34], Teflon [18], or other polymers, etc., have been actively developed recently. In addition, there have been some studies on the properties of piezoelectric balance (quartz plate) resonant systems, which oscillate near resonant base frequencies and thus increase microweighing effectiveness [35], as well as publications on the development of effective electrical excitation circuits for quartz plates within the scope of improving their resolution [36].

Previously, Mindlin's model was proposed in order to consider the heterogeneity of the junction in between the crystal plate and the coating layer (film) [37]: new methods of electrophysical and circuit design of electroacoustic transducers based on a single piezoelement were developed [38]. Besides, the search for new materials for manufacturing nanomechanical resonators is in continuous progress. In particular, a defect-free Au single-crystal nanowire (NWS) with resonant frequencies of 33–119 MHz has been shown to be a very promising material due to the almost absent energy scattering in the defect-free crystalline medium of the nanomaterial [39]. Other materials have been previously reported for piezoresonator development, among them lithium substrates such as lithium niobate ($LiNbO_3$) and lithium tantalate ($LiTaO_3$), aluminum nitride (AlN), and langasite ($La_3Ga_5SiO_{14}$) [15,22,23,40].

## 2.3. Coating Selection

A distinctive feature of selective mass-sensitive resonators (piezosensors) is the presence of a sorption coating characterized by differently pronounced selectivity with respect to the sorption—desorption of components from the near-electrode space (Figure 2A). At present, various phases, materials, native concentrates (extracts from natural objects and their solutions), polymer films, nanostructures, etc., are used to modify the resonators. The most commonly employed coatings are stationary chromatographic phases, from nonpolar (squalene, bee wax) to highly polar (polyethylene glycols and their esters) and specific ones (chemisorption coatings, dyes, polymer matrices, and polymers with DNA) [1]. Recently, plenty of new composite materials have been reported, among them $SnO_2$/CuO-based coatings for hydrogen sulfide vapor detection [41]; $NiCl_2$ and AgCl films deposited on nickel and silver electrodes of piezoquartz plates for ammonia detection [42]; nanoporous $TiO_2$ fibers functionalized with polyethylene diamine for formaldehyde vapors analysis [43]; titanate sol–gel layers imprinted with carbonic acids for highly sensitive detection of $C_2$–$C_4$ alcohols and $C_6$–$C_{10}$ hydrocarbons of normal and isomeric structures [44]; porphyrins and their derivatives [45–48]; ZnO nanoparticles modified with chiral porphyrin derivatives for enantiomer vapor recognition [49]; copolymers of porphyrin-substituted polypyrrole and carboxylated single-wall carbon nanotubes (SWCNTs) for volatile organic compounds (VOCs), *n*-butanol, in particular [50].

Molecularly imprinted polymers (MIP) are widely used to determine the variety of biomolecules [51–53]. Thus, polymeric films impregnated with birch and nettle pollen have been developed for the detection of allergens in the air [54]. The use of self-assembling glucosamine monolayers as biomimetic receptors for influenza viruses and other biomolecules has been reported in [55]. MIP-based QCM sensors were developed for the detection of low- and high-density lipoproteins (LDLs and HDLs) and were satisfactorily tested in human serum [56,57].

Thin films of polyaniline and emeraldine have been employed for assessments of the primary aliphatic alcohols methanol, ethanol, 2-propanol, and 1-propanol [58]; formic acid [59]; and amine vapors [60]. In the last case, the sensor's selectivity was based on differences in the values of diffusion coefficients and the kinetics of vapor adsorption of various amines. A sensor for methane with a detection limit of 0.15 vol % on the basis of supramolecular kriptophan A synthesized from vanillin alcohol was developed in Reference [61]. Sensitive film coatings made of fulvic acids isolated from humus [62], Langmuir–Blodgett calix [4] resorcinarene [63], and doped and nondoped multilayer carbon nanotubes (CNTs) [64] have been reported for toluene and several other VOCs detection (among them: benzene, acetone, ethanol hexane, cumene, ethylbenzene, ethyl acetate, ethanol). Despite the diversity of materials and approaches for the formation of sensitive coatings on the electrodes of the piezoelectric element, the reproducibility of coating properties determines the main performance characteristics of piezoquartz microbalances, and depending on the nature of the sensitive coating, the time of stable operation ("lifetime") of sensors varies from several days to a year. At the same time, in many cases, special attention is paid to the "training" of sensors before use and the exposure to pairs of high-concentration test compounds to form a stable structure. However, obtaining commercial sensors for long-term operation remains a challenging task in modern sensorics.

## 3. Single Piezosensor Applications

High mass sensitivity and fast response times in a piezoelectric resonator allow for active applications of piezoquartz microweighing in different fields [1]. Thus, a lot of attention is devoted to the selective detection of several species in air, among them carbon and nitrogen oxides [65], formaldehyde, ammonia [58,66], VOC–arenes [52,58,62], alcohols, ketones [50,58,66], amines [67,68], pesticides [69,70], short-chain aliphatic acids [59,71], etc. Moreover, often it is important to detect these compounds in a sample separately, since most of them belong to marker substances for assessing either environmental safety (nitrogen oxides, formaldehyde, arenas, pesticides) or the state of living objects (alcohols, ketones, amines, ammonia, acids (for all types of bioassays and food systems).

Atmospheric monitoring comprises not only the determination of hazardous and toxic compounds, but also the estimation of the size and the concentration of aerosol particles. For this purpose, in Reference [72], QCM sensors were employed to measure hourly and daily variations in the size and concentration of aerosol particles in the surface zone of semiarid rural places in India with low relative humidity (less than 75%). The use of a miniaturized measuring device with a fast response time allowed for obtaining valuable results for modeling the state and evolution processes of aerosols. In Reference [73], QCM sensors were employed for the direct measurement of stearic acid and homolog film mass changes during their dissolution process upon contact with solutions of anionic and non-ionic detergents. The use of QCMs made it possible to isolate and quantitatively characterize a series of successive stages (the adsorption and absorption of water and detergents on the film surface): a relationship was established between the film removal time and the detergent concentration in the solution [73]. In Reference [74], an atrazine sensor based on a QCM with a molecularly imprinted film of titanium dioxide was developed.

Piezoquartz microbalances are still an indispensable tool to study thin polymer layer behavior, in particular the changes in their mechanical and viscoelastic properties in comparison to bulk samples [75], and to investigate adsorption processes kinetics, for instance in the case of polyvinyl imidazole adsorption on a copper electrode [76]. The use of piezoresonators based on high-frequency

quartz plates (more than 10–15 MHz) with a single-sided coating is recommended as a universal tool for determining the dry residue in natural and drinking water, food, fuel, etc.

## 4. Arrays of Piezosensors as Measuring Elements of Multicomponent Detectors

In the last two decades, artificial sensing systems (e-nose and e-tongue) have been actively used to solve a wide range of tasks in order to obtain information on the integral characteristics of multicomponent samples [9,10,77]. Among the most common applications of e-nose are the analysis of light and medium volatile odor fractions of various food products for quality assessment [78,79], safety assessments of plastics and building materials [80,81], quality monitoring of motor oils and diesel fuels [44,82], medical diagnostics [45–48,83], and environmental monitoring [5,12,42].

### 4.1. Foodstuffs Analysis

Research on the use of chemical sensors for food analysis is actively developing. This is proven by the nine-fold increase in publications over the past 15 years (Figure 3) and the emergence of new research directions. Sensors have been employed in foodstuffs analyses both in single mode or inside e-nose and e-tongue systems, which in turn were configured to detect trace amounts of specific markers and predict the state of tested objects after prior training on test compounds. An analysis of publications shows that the choice of specific markers of the state of various systems obeys the following principle: normal (native state of systems) → possible destructive processes (damage, microorganism activity) → products of these processes (specific, often gaseous markers). An application of e-nose systems for foodstuffs became a very attractive technique due to the analysis simplicity, the minimal sample preparation required, and the proximity of the approach in odor assessments to organoleptic testing (Figure 4).

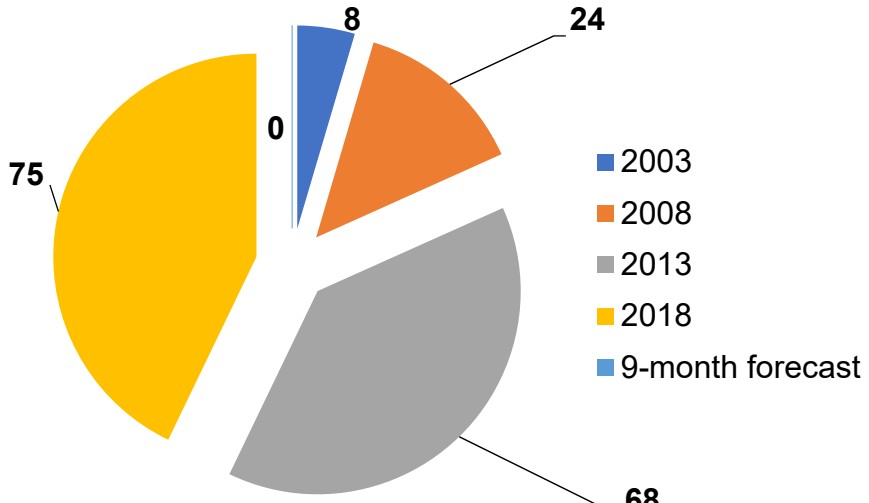

**Figure 3.** Dynamics of changes in the number of publications in food analysis employing chemical gas sensors.

Among the most common applications of e-nose systems is the continuous control of technological processes. For instance, in Reference [84], an array of a small number of piezosensors (six elements with MIP coatings) was employed to establish the characteristics of the composts obtained from grass and pine. The possibility of detecting a large number of alcohols and terpenes was shown, and the possibility of industrial monitoring of such processes using artificial sensing systems was highly appreciated. Among other applications of e-nose systems are the optimization of grape drying times in the production of sweet wines [85], the prediction of Fuji apple storage time [86], aroma studies of white pepper with various genotypes [87], vinegar component identification [88], flavor control of chocolate

products [89], ethanol concentration evaluations in Chinese spirits [90], fuel quality assessments based on an evaluation of oxidation product accumulation [44], food allergens detection [78,91], etc.

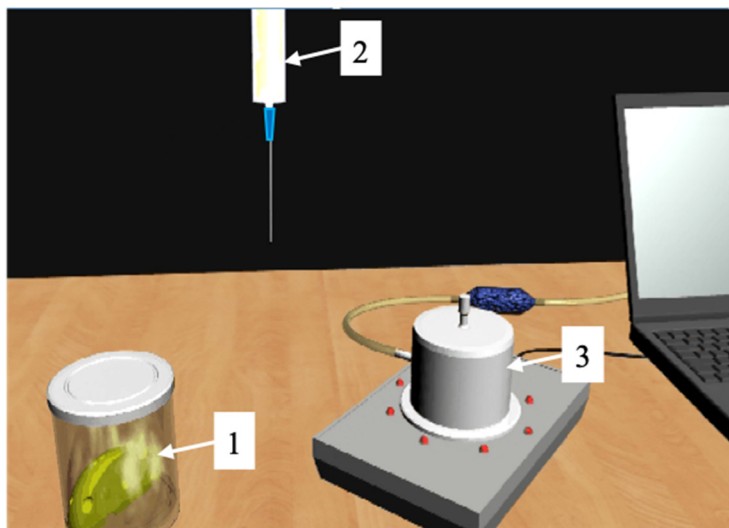

**Figure 4.** 3D model of food analysis system. 1: A sample in a sampler with an equal gas phase (EGP) above it; 2: syringe for EGP sampling and for injection into the e-nose detector; 3–8: sensor e-nose.

A great number of publications have been devoted to the development of industrial measuring metal oxide sensor-based systems for product identification and freshness control. Furthermore, chemometric methods of data processing are widely used to increase analysis informativeness and offer various approaches to sample preparation before analysis [10]. The integration of different types of sensors into an array and its subsequent alliance with gas chromatography may solve the task with a reliable determination of the content of volatile compounds that in part form an odor. Consequently, the results of studies have coincided with the results of arbitration methods and panelist tasting. Thus, the reliability of the results and the informativeness of the e-nose systems have been proven by traditional organoleptic assessments of products such as beer [92], oysters [93], shrimp [94], vinegar [88], and milk [95]. According to the results, organoleptic evaluation and e-nose systems showed high correlations in the sample rankings, additionally allowing for the identification of the substances responsible for sample differentiation [92].

Additional properties may be acquired by the combination of mass-sensitive QCM-based e-noses with other techniques, in particular mass spectrometry [92]; microbiological analysis [93]; standard physicochemical quality indicators [95]; and gas chromatography combined with mass spectrometry (GC/MS) for the classification of rice wine, vinegar [88], VOCs [83,96], and various terpenes of basil and mint leaves [74]. Chemometric methods are widely used to process data from arrays of measuring elements: the most common techniques employed are principal component analysis (PCA), cluster analysis, linear discriminant analysis (LDA), various regression methods, and artificial neural networks (ANNs) [9,10].

### 4.2. Environmental Monitoring

An application of chemical sensors and sensor arrays is one of the most promising opportunities to carry out inexpensive and real-time environmental monitoring [1,5,6,9,10]. Previously, several research works have been conducted on the application of piezoelectric sensors in assessments of natural and wastewater pollutants for quality control of the construction and finishing materials. Thus, using selective piezobalance sensors, the possibility of detecting profenofos in the concentration ranged from $10^{-8}$ to $10^{-5}$ mg/L [51], and atrazine with a detection limit of 0.1 µg [74] in wastewater was shown. The quantitative and kinetic parameters of the sorption of volatile compounds evaporating from the surface of self-adhesive film, construction putty, and polymer plates were established with

BAW microbalances in References [44,80]. A test method has been developed for evaluating the "background" of building materials in indoor air using signals from piezosensors in a matrix [80,81]. The developed approach has been applied to study the destruction of biodegradable polymer films and to find the optimal formulation. In Table 1, the applications of single piezosensors and multisensory systems employed for environmental monitoring and foodstuffs analysis are summarized.

**Table 1.** Summary of single piezosensors and multisensory system applications for environmental monitoring and foodstuffs analysis.

| Analysis Objects | Technique | Sensor Coating Type | Measuring System | Ref. |
|---|---|---|---|---|
| $CO_2$ gas | SAW | Teflon | single sensor | [18] |
| proteins and intact microorganisms | FPW | surface-immobilized coatings of biotinylated anti-dog IgG and dog-IgG antibodies | chemical sensor arrays with differential outputs | [21] |
| cell adhesion, cytotoxicity, cell viability, cell phenomena | QCM-D | cell preincubated QCM-D sensors | $\Delta D$–$\Delta f$ measurements in the presence of different reagents | [30] |
| hemagglutinin (HA) glycoprotein of influenza virions (H5N1) | QCM | polyepitope-functionalized Au NPs | antibody–antigen binding-based gravimetric immunosensor | [31] |
| dopamine (DA) | QCM-D | chitosan nanoparticles (CSNPs) | piezoelectric sensors array with crown ether coatings | [32] |
| aflatoxin B1 | QCM | biomolecule-functionalized magnetic nanoparticles | immunoassay | [33] |
| different proteins using recombinant DNA technology | QCM | silica particles induced by the GFP-R1 chimeric protein | single immunosensor | [34] |
| $H_2S$ vapor | SAW | $SnO_2$/CuO coatings | single sensor | [41] |
| $NH_3$ | QCM | $NiCl_2$ and AgCl films | single sensor | [42,66] |
| formaldehyde | QCM | nanoporous $TiO_2$ fibers | single sensor | [43] |
| VOCs: ethanol, *n*-propanol, *n*-butanol, *n*-hexane, *n*-heptane, *n*-/iso-octane, *n*-decane and monitoring emanation of degraded engine oil | QCM | titanate sol–gel layers imprinted with carbonic acids | single sensor | [44] |
| VOC cancer markers | QCM | porphyrins and their metallic complexes | e-nose composed from 8–12-element chemical sensor array | [45–48] |
| chiral VOCs | QCM | porphyrin–ZnO nanoparticle conjugates | single sensor | [49] |
| *n*-butanol | QCM | electropolymerized porphyrin-containing coating | single sensor | [50] |
| profenofos | QCM | MIP based on 11-mercaptoundecanoic acid (MUA) | single sensor | [51] |
| immunoglobulin G (IgG) | QCM | MIP polydopamine films | single sensor | [52] |
| wheat germ agglutinin (WGA) lectin | QCM | MIP polyacrylic film | single sensor | [53] |
| pollen allergens (birch and nettle) | QCM | pollen-imprinted polyurethanes | single sensor | [54] |
| influenza viruses and other biomolecules | QCM | self-assembling glucosamine monolayers | single sensor | [55] |
| low-density (LDLs) and high-density (HDLs) lipoproteins in blood serum | QCM | MIPs from acrylic acid (AA), methacrylic acid (MAA), and *N*-vinylpyrrolidone (VP) monomers in different ratios | single sensors | [56,57] |
| methanol, ethanol, 2-propanol, and 1-propanol vapors | QCM | thin polyaniline film | single sensor | [58] |
| formic acid gas | QCM | thin polyaniline film | single sensor | [59] |

**Table 1.** *Cont.*

| Analysis Objects | Technique | Sensor Coating Type | Measuring System | Ref. |
|---|---|---|---|---|
| aliphatic amine vapors | QCM | polyaniline/emeraldine base (PANI/EB) film | single sensor | [60] |
| $CH_4$ gas | QCM | supramolecular cryptophane-A film deposited via electrospray method | single sensor | [61] |
| toluene | QCM | sensitive film coatings made of fulvic acids isolated from humus | single sensor | [62] |
| VOCs: ethanol, benzene, toluene, ethylbenzene, ethyl acetate, acetone, hexane, and cumene | QCM | Langmuir–Blodgett calix [4] resorcinarene | single sensor | [63] |
| VOCs: benzene, methylbenzene, 1,2-dimethylbenzene, ethylbenzene, isopropylbenzene, 1,2,4-trimethylbenzene, monoatomic aliphatic alcohols ($C_2$—$C_9$) with normal and isomeric structures | QCM | doped and nondoped multilayer carbon nanotubes (CNTs) | MAG-8 gas analyzer using e-nose methodology with an array of eight piezoelectric sensors | [64] |
| simultaneous detection of $CO_2$ and $NO_2$ | SAW | $CO_2$-sensitive film (Teflon AF 2400) and an $NO_2$-sensitive film (indium tin oxide) | multigas sensor | [65] |
| alkylamines | QCM | polymer and solid-state thin films, thin films of acid–base indicators | single sensor, gas analyzer | [67,68] |
| organophosphorus and carbamate pesticides | QCM | acetylcholinesterase (AChE) immobilized on QCM surface | single sensor | [69] |
| chain aliphatic acids | QCM | standard chromatographic coatings: polyethylene glycol PEG-2000 (PEG); the ethers: polyethylene glycol adipate (PEGA), phthalate (PEGP), etc.; the specific sorbents 18-crown-6 (18C6) and propolis (Pr) | MAG-8 analyzer using e-nose methodology with an array of eight piezoelectric sensors | [71] |
| variations in the size and concentration of aerosol particles | QCM | no coating | single sensor | [72] |
| stearic acid and homolog dissolution monitoring | QCM | no coating | single sensor | [73] |
| atrazine sensor | QCM | molecularly imprinted film of titanium dioxide | single sensor | [74] |
| adsorption of poly(vinylimidazole) (PVI) on Cu | QCM | no coating | single sensor | [76] |
| shrimp allergen determination in food | QCM | self-assembly of 1,6-hexanedithiol (HDT) and antishrimp antibodies | e-nose immunosensor system | [79] |
| monitoring terpene emissions from odoriferous plants | QCM | molecularly imprinted polymer (MIP) selectively interacting with alpha-pinene, thymol, estragol, linalool, and camphor | sensor array | [80] |
| free volatile components from phenolformaldehyde plastics | QCM | nonspecified | sensor array | [81] |
| composts of grass and pine characterization, alcohol and terpene detection | QCM | affinity materials and MIPs | sensor array of six piezosensors | [84] |
| volatile food flagrancies in confectionary masses | QCM | nonspecified | sensor array | [89] |
| organoleptic indicators of milk assessments | QCM | nonspecified | sensor array | [95] |
| wastewater quality assessments | QCM | MWCNTs, zirconium(IV) oxynitrate, biohydroxyapatite coatings | MAG-8 gas analyzer with six piezosensors | This work's case study |

Below, a case study devoted to the recent application of a multiselective QCM array for the assessment of wastewater quality is discussed in detail in order to illustrate the ability of such a multisensor system (based on piezosensors) to address real-world analytical tasks.

### 4.3. Case Study: QCM E-Nose for Wastewater Quality Assessment

Tests were performed on 15 wastewater samples provided by the Quality Control Laboratory of the LCEC (Lipetsk City Energy Company) and were analyzed with a gas analyzer MAG-8 comprised of six multiselective piezosensors. The sensors were modified by nanomaterials of various natures and coating weights, thus resulting in the additional selectivity of analyte sorption from the air environment above the wastewater solutions (Table 2). The statistically processed responses of the MAG-8 device in the analyzed samples are presented in Table 3. These responses were correlated to wastewater indicators obtained via standard methods (more than 40 controlled indicators), and some of them are listed in Table 4.

The main task of the experiment was to establish the possibility of wastewater sample assessment with a piezoelectric MAG-8 e-nose as well as the correlation of the e-nose response with water quality indicators determined by standard methods. A principal component analysis (PCA) was employed in order to compare the data obtained by the gas analyzer (Table 3) to the results obtained through the standard methods and Table 4 and to find out the correlation between these two datasets. The PCA method provides the identification of tested samples without prior training and model building. For the PCA analysis, wastewater samples were chosen as specimens, and e-nose output data containing qualitative (sorption effectiveness parameters [97], $A_{ij}$) and quantitative (analytical sensors signal, $\Delta F_i$; visual prints area, $S_i$) information were selected as variables. All data were autoscaled in order to reduce the impact of individual variables on the modeling results.

It was established that the use of three principal components described 90% of the data: the most important were the first two principal components, representing (respectively) 36% and 24% of the overall system variance, as seen in Figure 5. On the scores chart (Figure 5A), two separate sample groups can be distinguished: the first one was formed by samples #04, #05, and #51, which were the most different from the rest of the analyzed samples; while samples #47, #02, and #41, the classification of which was virtually not influenced by the selected variables, constituted the second group. When comparing scores and loadings charts (Figure 5), it was established that the quantitative output data—mainly the "visual fingerprints" areas—had the greatest impact on the identification of samples #04, #05, and #51. The output data for the first sensor had the greatest impact on the discrimination of samples #39, #40. #42, and #49, and this result indicated the presence of volatiles (organic trace impurities containing oxygen) with a high affinity to the oxidized MWCNT coating of Sensor 1.

**Table 2.** The coating compositions and weights of QCM sensors in the MAG-8 gas analyzer.

| Sensor | Coating | Coating Weight (μg) |
| --- | --- | --- |
| Sensor 1 | Multiwalled carbon nanotubes (MWCNTs) oxidized by nitric acid | 5.03 |
| Sensor 2 | Zirconium(IV) oxynitrate | 4.03 |
| Sensor 3 | Biohydroxyapatite | 4.03 |
| Sensor 4 | Biohydroxyapatite | 2.15 |
| Sensor 5 | Zirconium(IV) oxynitrate | 2.12 |
| Sensor 6 | Multiwalled carbon nanotubes (MWCNTs) oxidized by nitric acid | 1.96 |

**Table 3.** The response of the MAG-8 e-nose in wastewater samples.

| Sample # | $\Delta F_i$ | Sensor 1 | Sensor 2 | Sensor 3 | Sensor 4 | Sensor 5 | Sensor 6 | $S_i$ (Hz/s) |
|---|---|---|---|---|---|---|---|---|
| Potable Water | **X ± ΔX** | 6 ± 1 | 7 ± 1 | 12 ± 1 | 7 ± 1 | 6 ± 1 | 11 ± 1 | 125 ± 20 |
| | **Δ** | 0.23 | 0.19 | 0.12 | 0.21 | 0.23 | 0.21 | 0.16 |
| 04 | **X ± ΔX** | 12 ± 2 | 14 ± 2 | 25 ± 2 | 12 ± 2 | 12 ± 1 | 20 ± 2 | 485 ± 30 |
| | **Δ** | 0.23 | 0.20 | 0.09 | 0.22 | 0.11 | 0.13 | 0.08 |
| 01 | **X ± ΔX** | 13 ± 1 | 15 ± 1 | 26 ± 1 | 15 ± 2 | 12 ± 1 | 24 ± 2 | 480 ± 40 |
| | **Δ** | 0.11 | 0.09 | 0.05 | 0.16 | 0.12 | 0.10 | 0.08 |
| 51 | **X ± ΔX** | 12 ± 1 | 15 ± 2 | 24 ± 2 | 15 ± 2 | 12 ± 1 | 21 ± 2 | 470 ± 50 |
| | **Δ** | 0.12 | 0.19 | 0.12 | 0.19 | 0.12 | 0.12 | 0.14 |
| 05 | **X ± ΔX** | 12 ± 1 | 15 ± 1 | 24 ± 1 | 15 ± 2 | 12 ± 1 | 21 ± 2 | 445 ± 30 |
| | **Δ** | 0.12 | 0.09 | 0.06 | 0.16 | 0.12 | 0.13 | 0.09 |
| 47 | **X ± ΔX** | 12 ± 1 | 14 ± 1 | 25 ± 2 | 14 ± 1 | 11 ± 1 | 19 ± 1 | 420 ± 20 |
| | **Δ** | 0.12 | 0.10 | 0.12 | 0.10 | 0.13 | 0.07 | 0.06 |
| 49 | **X ± ΔX** | 12 ± 1 | 14 ± 1 | 24 ± 1 | 14 ± 1 | 12 ± 2 | 19 ± 1 | 430 ± 30 |
| | **Δ** | 0.12 | 0.01 | 0.06 | 0.01 | 0.2 | 0.07 | 0.08 |
| 48 | **X ± ΔX** | 11 ± 1 | 12 ± 1 | 24 ± 1 | 13 ± 2 | 11 ± 1 | 18 ± 2 | 420 ± 20 |
| | **Δ** | 0.13 | 0.12 | 0.06 | 0.19 | 0.13 | 0.14 | 0.05 |
| 50 | **X ± ΔX** | 12 ± 1 | 14 ± 1 | 24 ± 1 | 14 ± 1 | 11 ± 1 | 19 ± 1 | 435 ± 20 |
| | **Δ** | 0.13 | 0.10 | 0.06 | 0.10 | 0.13 | 0.07 | 0.04 |
| 52 | **X ± ΔX** | 10 ± 1 | 14 ± 1 | 24 ± 1 | 13 ± 1 | 12 ± 1 | 19 ± 1 | 350 ± 20 |
| | **Δ** | 0.15 | 0.10 | 0.06 | 0.10 | 0.13 | 0.07 | 0.07 |
| 02 | **X ± ΔX** | 10 ± 1 | 14 ± 1 | 24 ± 1 | 14 ± 1 | 11 ± 1 | 19 ± 1 | 420 ± 20 |
| | **Δ** | 0.01 | 0.10 | 0.06 | 0.10 | 0.13 | 0.13 | 0.08 |
| 03 | **X ± ΔX** | 13 ± 1 | 14 ± 1 | 23 ± 1 | 13 ± 1 | 11 ± 1 | 19 ± 1 | 440 ± 20 |
| | **Δ** | 0.11 | 0.10 | 0.06 | 0.10 | 0.03 | 0.03 | 0.06 |
| 39 | **X ± ΔX** | 14 ± 1 | 14 ± 1 | 24 ± 1 | 14 ± 1 | 11 ± 1 | 19 ± 1 | 460 ± 30 |
| | **Δ** | 0.10 | 0.10 | 0.06 | 0.10 | 0.13 | 0.07 | 0.08 |
| 40 | **X ± ΔX** | 14 ± 1 | 14 ± 1 | 23 ± 1 | 13 ± 1 | 11 ± 1 | 19 ± 1 | 420 ± 20 |
| | **Δ** | 0.10 | 0.01 | 0.06 | 0.10 | 0.01 | 0.07 | 0.06 |
| 42 | **X ± ΔX** | 13 ± 1 | 14 ± 1 | 23 ± 1 | 13 ± 1 | 11 ± 1 | 19 ± 1 | 400 ± 30 |
| | **Δ** | 0.21 | 0.10 | 0.06 | 0.10 | 0.01 | 0.08 | 0.08 |
| 41 | **X ± ΔX** | 12 ± 1 | 14 ± 1 | 23 ± 1 | 15 ± 1 | 11 ± 1 | 20 ± 2 | 490 ± 20 |
| | **Δ** | 0.12 | 0.10 | 0.06 | 0.09 | 0.13 | 0.15 | 0.04 |

**Table 4.** The results of several different metal concentrations (C, μg/L) in wastewater samples (determined by standard methods).

| Sample # | Mn | Al | Pb | Cr | Fe | Ni | Zn | Cu | Cd |
|---|---|---|---|---|---|---|---|---|---|
| 01 | 0.062 | 0.086 | 0.0045 | <0.01 | 9.42 | 0.0031 | 0.09 | 0.0089 | <0.0001 |
| 02 | 0.04 | 0.20 | <0.001 | <0.01 | 0.61 | 0.0016 | 0.12 | 0.0097 | <0.0001 |
| 03 | 0.52 | 2,6 | 0.520 | 0.11 | 52 | 0.018 | 0.8 | 0.60 | 0.0027 |
| 04 | 0.102 | 0.64 | <0.001 | <0.01 | 2.4 | 0.0029 | 0.12 | 0.034 | <0.0001 |
| 05 | 0.181 | 2.24 | - | <0.01 | 6.8 | 0.0096 | 0.55 | 0.10 | 0.0004 |
| 39 | 0.095 | 0.56 | 0.0022 | 0.039 | 1.0 | 0.0049 | 0.16 | 0.031 | 0.0002 |
| 40 | 0.16 | 1.46 | 0.0047 | 0.012 | 1.2 | 0.0033 | 0.12 | 0.040 | 0.0002 |
| 41 | 0.092 | 0.24 | 0.0120 | <0.01 | 1.8 | 0.0048 | 0.18 | 0.050 | 0.0015 |
| 42 | 0.17 | 0.63 | 0.0033 | 0.013 | 2.8 | 0.0044 | 0.43 | 0.108 | 0.0003 |
| 47 | 0.118 | 2.12 | 0.011 | 0.025 | 4.4 | 0.030 | 0.33 | 0.096 | 0.0003 |
| 48 | 0.098 | 1.14 | 0.009 | <0.01 | 4.2 | 0.0058 | 0.98 | 0.036 | <0.0001 |
| 49 | 0.17 | 0.33 | 0.0041 | <0.01 | 1.9 | 0.011 | 0.20 | 0.048 | <0.0001 |
| 50 | 0.096 | 0.32 | 0.0029 | <0.01 | 2.7 | 0.061 | 0.11 | 0.053 | 0.0002 |
| 51 | 0.16 | 0.25 | 0.0014 | <0.01 | 0.43 | 0.0016 | 0.013 | 0.0055 | <0.0001 |
| 52 | 0.126 | 0.148 | 0.0026 | <0.01 | 0.26 | 0.0014 | 0.011 | 0.004 | <0.0001 |

From the loadings chart (Figure 5B), it can be seen that the qualitative parameters ($A_{ij}$) for Sensor 1, Sensor 2, and Sensor 6 were grouped separately, which indicates the significant influence of these sensors on sample classification. This fact can be explained by the active sorption of O- and N-containing compounds and water vapors on sensors with polar MWCNTs of various weights and zirconium(IV) oxynitrate coatings.

Similarly, a PCA procedure was carried out for a data array of water quality indicators determined by standard methods, as seen in Figure 6. As can be seen from the score plot (Figure 6A), most of the samples were placed in the graph center, which means that the standard indicators were insufficiently informative for water sample classification. The PCA procedure identified samples #01 and #47 as outliers, since these samples were situated far from the main group of other water samples on the PCA score plot, thus representing abnormal values of the measured standard indicators.

From the loadings graph (Figure 6B), it can be established that the "suspended solids" and "anion-active surfactants" variables were the most discriminative along the first principal component, PC1, while the "petroleum products" content and the "hydrogen index" were the most influencing variables along axis PC2. Moreover, the water mineral composition indicators were correlated with each other and did not influence the sample classification.

When comparing the PCA models of water sample classifications obtained from piezoelectric e-nose output data and from standard quality indicators (Figures 5 and 6, respectively), it was apparent that the samples ranged differently, which made it impossible to establish interconnections between the two datasets. Thus, we calculated the Pearson paired criterion to evaluate the interconnection between the sensor array data and the standards of water quality parameters, as seen in Table 5.

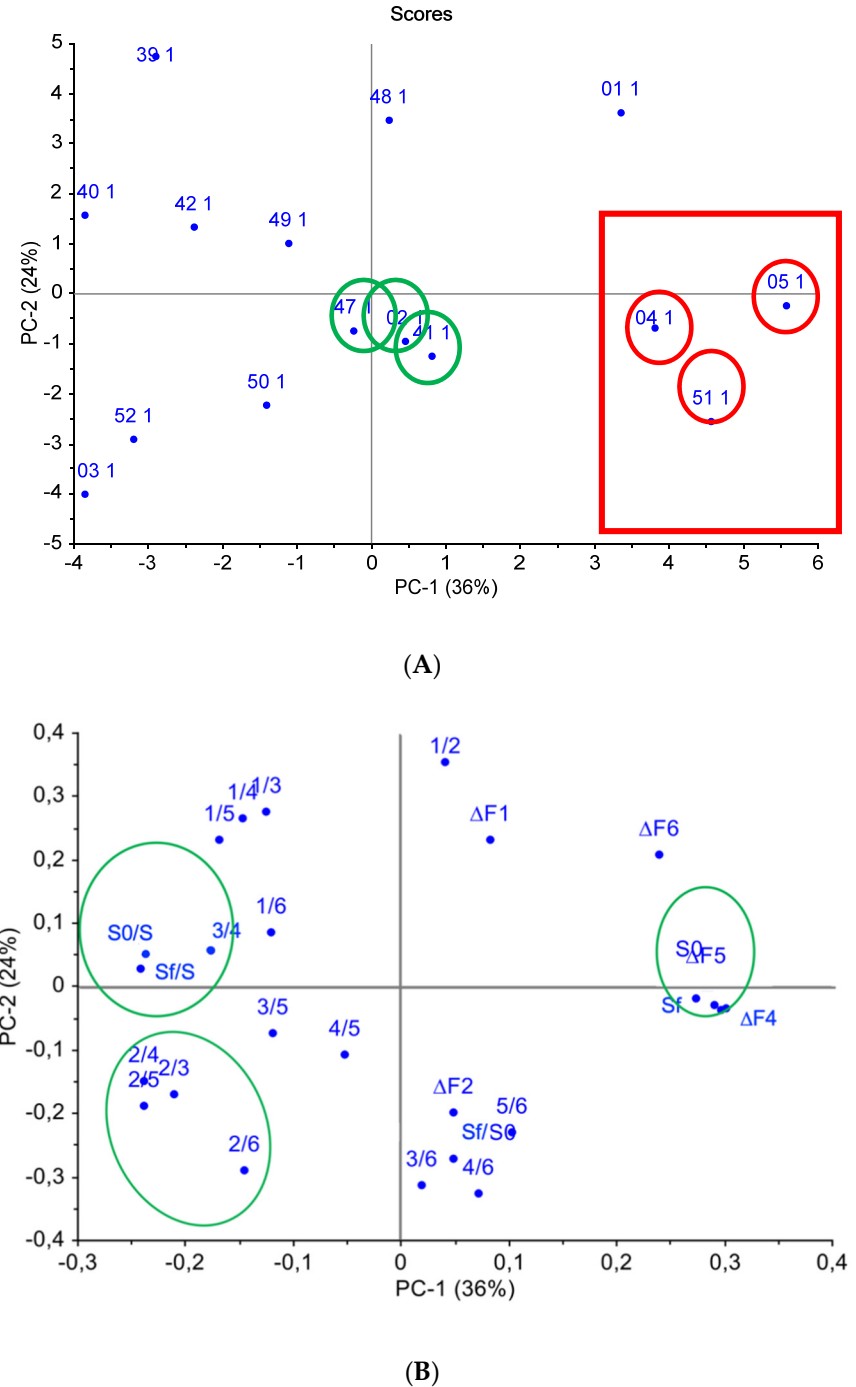

**Figure 5.** Principal component analysis (PCA) of water samples according to sensor array output data: (**A**) scores chart, (**B**) loadings chart.

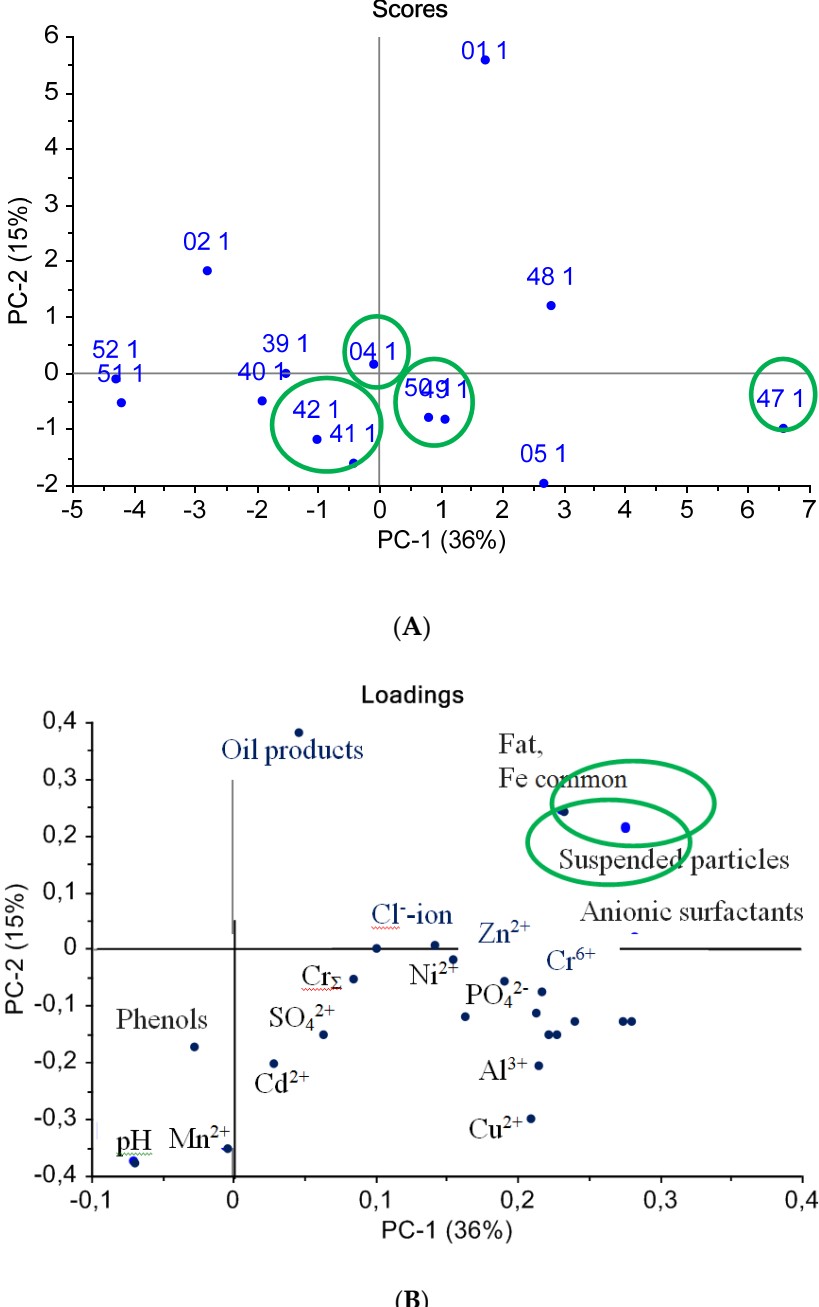

**Figure 6.** PCA analysis of water samples using standard methods: (**A**) scores chart, (**B**) loadings chart.

**Table 5.** Pearson correlation coefficients between e-nose output data and the standard parameters of water quality.

| Parameters | $A_{13}$ | $A_{14}$ | $A_{15}$ * | $A_{16}$ | $A_{23}$ | $A_{24}$ | $A_{25}$ | $A_{26}$ | $A_{36}$ | $A_{45}$ | $A_{56}$ |
|---|---|---|---|---|---|---|---|---|---|---|---|
| $Co^{2+}$ | 0.595 | 0.509 | - | - | - | - | - | - | - | - | - |
| $Cr^{3+}$ | 0.536 | 0.474 | - | 0.442 | - | - | - | - | - | - | - |
| $Cr_{total}$ | - | - | - | - | 0.496 | 0.456 | 0.493 | 0.434 | - | - | - |
| $Cd^{2+}$ | - | - | - | - | 0.507 | 0.453 | - | 0.436 | - | - | - |
| $V^{3+}$ | - | - | - | - | - | - | 0.757 | 0.529 | - | - | - |
| $Cu^{2+}$ | - | - | - | - | - | - | 0.409 | 0.433 | - | - | - |
| $As^{3+}$ | - | - | - | - | - | - | 0.435 | 0.409 | - | - | - |
| $Pb^{2+}$ | - | - | - | - | - | - | 0.426 | 0.405 | - | - | - |
| $Mn^{2+}$ | - | - | - | - | - | - | - | 0.459 | - | - | - |
| Phosphates (P) | 0.478 | 0.597 | - | - | - | - | - | - | - | - | - |
| $S^{2-}$ | - | - | - | - | 0.473 | - | - | 0.482 | - | - | - |
| $F^-$ | - | - | - | - | - | - | - | - | 0.797 | - | - |
| Fats | - | - | - | - | - | - | - | - | - | - | 0.422 |
| Phenols | - | - | - | - | - | - | - | - | - | 0.371 | - |
| Oil products | - | - | - | - | - | - | 0.422 | - | - | - | - |

* Correlation coefficients are statistically insignificant.

Table 5 demonstrates only the statistically significant correlation coefficients obtained. It can be seen that no significant correlation was established between the sensor array outputs (analytical signals of sensors and "visual prints" area) and the standard water quality indicators. On the contrary, the most informative sensor array outputs were the parameters of the efficiency of sorption ($A_{13}$, $A_{14}$, $A_{25}$, $A_{26}$), which were associated with the mineral compositions of the analyzed samples. It is known that the presence of dissolved salts changes the pressure of saturated water vapor and other dissolved volatile substances, and these changes were then registered by sensors with polar coatings.

Indicators such as "suspended solids" and "hydrogen index" did not have a statistically significant relationship with any output parameter of the sensor array. This finding permitted a significantly simplified sample pretreatment prior to the analysis of any objects containing suspensions. In fact, it was demonstrated by means of the paired correlations method in wastewater that the "suspended solids" index and the pH of the solution did not affect the response of e-nose sensors.

We then estimated the possibility of predicting the value of standard indicators of water samples from e-nose output data. The multilinear regression method (MLR) was employed for this purpose. For each standard indicator, the MLR model was first constructed and then fitted by the gradual elimination of insignificant variables: finally, the model adequacy was estimated. For the majority of standard indicators, it was not possible to obtain significant MLR models based on sensor array output; however, adequate MLR models were obtained for the evaluation of $Co^{2+}$, $Cd^{2+}$, $V^{3+}$, phosphate, and fat content. The MLR results are listed in Table 6.

As can be seen from Table 6, the most precise model (root mean squared error of prediction (*RMSEP*) = 0.0003) was obtained for the prediction of vanadium content in water: the most informative variables for this model construction were sorption effectiveness parameters for Sensor 1 (MWCNT-coated) and Sensor 2 ($ZrO(NO_3)_2$-coated). Due to the small dataset available, a one-leave-out cross-validation method was employed for the *RMSEP* calculation. The obtained *RMSEP* values varied within the 5%–25% range, with the smallest *RMSEP* being less than 2% for vanadium content.

The resulting MLR models could be considered for simultaneous evaluations of several parameters due to the nonspecific response of sensor coatings to substances whose content must be predicted. For example, it was possible to utilize the obtained MLR model for vanadium to estimate the content

of other heavy metal ions, use the MLR model for phosphates to assess the content of other anions, or to predict the content of oily substances using the MLR model for fats.

**Table 6.** Multiple linear regression (MLR) coefficients (β), their significance levels (*p*), and the root mean squared error of prediction (*RMSEP*) for standard water quality indicators.

| Projected Indicator | | $b_0$ | $A_{12}$ | $A_{13}$ | $A_{14}$ | $A_{23}$ | $A_{24}$ | $A_{25}$ | $A_{26}$ | $A_{34}$ | $A_{46}$ | *RMSEP* |
|---|---|---|---|---|---|---|---|---|---|---|---|---|
| $Co^{2+}$ | β | −0.034 | - * | 0.281 | −0.129 | - | – | - | - | - | - | 0.0027 |
| | *p* | 0.012 | - | 0.063 | 0.100 | - | - | - | - | - | - | |
| $Cd^{2+}$ | β | 0.036 | - | - | - | - | −0.040 | - | 0.060 | - | −0.055 | 0.0007 |
| | *p* | 0.056 | - | - | - | - | 0.05 | - | 0.030 | - | 0.037 | |
| $V^{3+}$ | β | 0.172 | - | - | - | - | −0.205 | 0.030 | 0.275 | - | −0.276 | 0.0003 |
| | *p* | 0.001 | - | - | - | - | 0.001 | 0.001 | 0.001 | - | 0.001 | |
| Phosphates | β | −26.7 | 20.7 | −193 | 131.8 | - | - | - | - | - | - | 2.12 |
| | *p* | 0.011 | 0.017 | 0.032 | 0.009 | - | - | - | - | - | - | |
| Fats | β | 5046 | - | - | - | −4266 | - | - | 3270 | −1415 | −3420 | 21.7 |
| | *p* | 0.008 | - | - | - | 0.010 | - | - | 0.012 | 0.013 | 0.007 | |

\* The parameter was not employed for modeling.

## 5. New Opportunities and Trends for Piezosensors Development

One of the promising directions in the development of novel piezosensors is the establishment of appropriate transducers, which may implement simultaneously several operations principles, thus resulting in the construction of an analytical multisystem [4,98–100]. Although publications on this topic are still rare, they are of great interest in solving issues of increasing the information content and overall dimensions of sensory systems. Thus, Kim et al. have reported the development of a surface plasmon resonance (SPR)/QCM sensor with simultaneous measurement of the quartz plate resonant frequency oscillation and laser beam resistance and the intensity of reflection [101]. The sensor was obtained by spraying titanium (5 nm) and gold (50 nm) substrates onto different sides of a quartz plate: developed SPR/QCM sensors are promising for microchip technology applications.

Continuous progress has been observed in applications of sensor arrays, and e-noses in particular, in the analysis of complex objects. In Reference [97], new identification parameters, $A_{ij}$, were calculated from the signals of individual sensors of the array and were employed for the identification of some components such as amines, ammonia, and acetic acid in unknown mixtures at a microconcentration level. This approach makes it possible to solve identification problems for complex mixtures without the use of chemometric data processing methods and can be successfully applied when programming microchips and creating data processing algorithms obtained by portable mobile systems based on an array of piezosensors.

Quite a large number of "e-noses" are currently commercially available instruments [102–106]. However, to date, there have been few portable models on the market for these products [1]. E-nose technology, along with near infrared (NIR) spectroscopy, ultrasound images, and computer colorimetry, belongs to nondestructive control methods employed for rapid screening of a large number of samples in animal husbandry to assess their quality [107].

Another type of oscillatory system, microcantilevers with biocompatible surfaces using natural materials, has been employed to create highly accurate and miniaturized systems with low detection limits for a wide range of analytes [108]. The development of microcantilever sensors and sensor arrays is a promising direction for the creation of environmentally friendly electronics [109], and the works in this area may be interpreted as research in the field of "green" sensorics.

An analysis of publication activity over the past 20 years in international peer-reviewed journals, in this and previous reviews [1,7,9–14,108], the number of abstracts at major conferences dedicated not only to sensorics, but also in the area technology and information, all this allows objectively assess the relevance and interest of the scientific community public consumers to development, production and

implementation for wide range of applications the analyzers based on chemical sensors of various principles, including piezo quartz microbalances.

**Author Contributions:** Investigation, methodology, and original draft preparation, T.A.K.; review and editing, L.B.L. All authors have read and approved the final manuscript.

**Funding:** This research received no external funding.

**Conflicts of Interest:** The authors declare no conflict of interest.

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
