# Peer review of "A Perspective on Recent Advances in Piezoelectric Chemical Sensors for Environmental Monitoring and Foodstuffs Analysis"

_chemosensors, doi:10.3390/chemosensors7030039_

Round 1

Reviewer 1 Report

This review paper gives an overview on the development and application of chemical sensors based on piezoelectric quartz resonator. The reviews are quite interested in the fields of piezoelectric based chemosensors. However, the paper gave more details on the applications but less description on the sensor devices. I suggest the authors add some summarizations and figures on the common principals and designs on the piezoelectric quartz devices, especially on these sensors based on mechanical oscillatory system mentioned in figure 1. After this revision I think the paper is suitable for publication.

Author Response

We would like to thank a reviewer for the appreciation of our work and for the valuable comments. We have modified the Figure 1 and have added in the manuscript text   the descriptions  on the common principals and designs on the piezoelectric quartz devices  based on mechanical oscillatory system mentioned in Figure 1. All the changes made are highlighted in yellow.

Reviewer 2 Report

Dear Editor:

The authors submit for evaluation the manuscript entitled "Recent advances in piezoelectric chemical sensors for environmental monitoring and foodstuffs analysis: a review". The authors indicate in the abstract that the results are centered in the last 20 years. The topic is of interest and deserves a review in the journal. However I consider that It is not adequate in its current form.

Some suggestions:

- Although the authors know the diverse technologies, it would be interesting including a brief description of the technologies for scientist working in sensors but not used to the technology

- The number of references is very limited

- A table sumarising theanalyte, the kind of balance, the kind of sensor, the sensor coating and the reference would be convenient for clarity

- Section 3.3, the case study, seems a research article, it is not clear to which reference belong or if it is new. Including extensive research results in a review is not usual. 

Author Response

The authors submit for evaluation the manuscript entitled "Recent advances in piezoelectric chemical sensors for environmental monitoring and foodstuffs analysis: a review". The authors indicate in the abstract that the results are centered in the last 20 years. The topic is of interest and deserves a review in the journal. However I consider that It is not adequate in its current form.

- Although the authors know the diverse technologies, it would be interesting including a brief description of the technologies for scientist working in sensors but not used to the technology

Thank you for your valuable comments. We fully agree with a consideration that the description of piezoelectric sensors technologies can be useful for scientists working in the field of sensors. Therefore we have introduced the brief description on construction  and short discussion on working principle theoretical background of reported sensors in the manuscript. For this the new subsection, 2.1. Design and operation principle, was inserted and the Figure 1 and Figure 2 were updated with more details and sensors schematic presentation and images.  All the changes made are highlighted in yellow.

- The number of references is very limited

In our work we have reported  the selected publications in the area of  chemical sensors  based on piezoelectric quartz resonators in for environmental monitoring and foodstuffs analysis. Of course, the number of works dedicated to this argument is huge and in continuous growth, and it is also briefly mentioned in section 4.1 and in Figure 3 of the manuscript. Despite of this, the main aim of our work was to give general considerations, illustrating them with  several concrete examples reported in literature. We hope, that in this form the paper can useful for a large number of scientists working in the area of piezoelectric sensors development and application; which on their turn may perform more targeted search on specific arguments of interest. 

A table summarizing the analyte, the kind of balance, the kind of sensor, the sensor coating and the reference would be convenient for clarity

According to the reviewer suggestion the Table 1, entitled “Sum-up of single piezo sensors and multisensory systems applications for environmental monitoring and foodstuffs analysis” has been introduced in the manuscript.  The table contains the description of analysis objects, employed technique, sensor coating type, measuring system composition and corresponding reference number. The numbers of other Tables were hence accurately controlled and modified, when required. All the changes made are highlighted in yellow.

- Section 3.3, the case study, seems a research article, it is not clear to which reference belong or if it is new. Including extensive research results in a review is not usual. 

Thanks to referee for this comment. Actually, the results reported in section 4.3 Case study: QCM e-nose for waste water quality assessment are the new unpublished results obtained in our laboratories very recently. It is right, the unusual paper organization format, comprising first the general description and previously published researches and followed by a practical example in the form of case study is employed in our manuscript. By this our intention was to illustrate our latest results in a context of the previous works performed in this area. In order to clarify the results provenience, in a Table 1 we have specified, that the reported in case study research is new. The corresponding changes were yellow highlighted. We hope that the revised version of the manuscript may meet your approval  and can be suitable for publication in Chemosensors.

Round 2

Reviewer 1 Report

This review paper gives an overview on the development and application of chemical sensors based on piezoelectric quartz resonator. The reviews are quite interested in the fields of piezoelectric based chemosensors. The revised version is much improved. I am satisfied with the supplements and I would like to recommend the publication of this paper.  

Author Response

Dear Reviewer, thank you very much for your valuable comments and suggestions. 
We hope that the revised version of our manuscript will fit for publication in Chemosensors.